# Retention time of lakes in the Larsemann Hills oasis, East Antarctica

Elena Shevnina[1], Ekaterina Kourzeneva[1], Yury Dvornikov[2], Irina Fedorova[3]

[1]Finnish Meteorological Institute, Helsinki, Finland.

[2]Department of Landscape Design and Sustainable Ecosystems, Agrarian-Technological Institute, RUDN University, Moscow, Russia

[3]Saint-Petersburg State University, St. Petersburg, Russia.

Correspondence to: Elena Shevnina (elena.shevnina@fmi.fi)

**Abstract.** This study provides first estimates of water transport time scale for five lakes located in the Larsemann Hills oasis (69º23´S, 76º20´E) in East Antarctica. We estimated lake retention time (LRT) as a ratio of lake volume to the inflow and outflow terms of a lake water balance equation. The LRT was evaluated for lakes of epiglacial and land-locked types, and it was assumed that these lakes are monomictic, with water exchange occurring during the warm season only. We used hydrological observations collected in 4 seasonal field campaigns to evaluate the LRT. For the epiglacial lakes Progress and Nella/Scandrett, the LRT was estimated at 12–13 and 4–5 years, respectively. For the land-locked lakes Stepped, Sarah Tarn and Reid, our results show a great difference in the LRT calculated from the outflow and inflow terms of the water balance equation. The LRTs for these lakes vary depending on the methods and errors inherent to them. We relied on the estimations from the outflow terms, since they are based on hydrological measurements with better quality. Lake Stepped exchanged water within less than 1.5 years. Lake Sarah Tarn and Lake Reid are endorheic ponds, with water loss mainly through evaporation. Their LRTs were estimated as 21–22 years and 8–9 years, respectively. To improve the LRT estimates, special hydrological observations are needed to monitor the lakes and streams during the warm season with a uniform observational program.

## 1 Introduction

On the continent of Antarctica, most of the water is frozen and deposited in the ice sheets, glaciers and permafrost. Climate warming enhances melting of the ice sheets and glaciers, and melted water accumulates in lakes and streams. The lakes appear on the surface of the continental ice sheet, at its contact with rocks and in local depressions in ice-free areas (oases). Antarctic lakes exist both under the ice sheet (subglacial type lakes) and on top of it (supraglacial type). Many lakes are located on the boundary between the rocks and continental/shelf ice sheets (epiglacial and epishelf types). In oases, lakes of the land-locked or closed basin type occupy local relief depressions (Govil et al., 2016; Hodgson, 2012).

In warm seasons, numerous supraglacial lakes appear on the surface of the continental ice sheet over its edges, in "blue ice" regions, and in the vicinity of rock islands or nunataks (Leppäranta et al., 2020; Bell et al., 2017). These lakes may be up to 80 km long, and accumulate large amounts of liquid water potentially affecting the ice discharge, ice calving and hydro-

stability of the continental ice sheet (Stokes et al., 2019). Lakes of the epiglacial type are situated at glacier edges, and melting of the glacier ice is the main source of water inflow into them. These lakes may be perennially frozen, or partially free of ice during the austral summer lasting from December to February. Land-locked lakes appear in local depressions after retreat of the continental ice sheet. Precipitation and melting of seasonal snow cover are two main sources of water

inflow for these lakes. Precipitation over the lake surface usually contributes insignificantly to the water inflow compared to the snow melting (Klokov, 1979). Small land-locked lakes are fully ice-free for a period of 2–3 months in summer. Big land-locked lakes can stay partially ice covered in summer, and a number of such lakes are found in the Schimacher oasis, Thalla Hills and Bunger Hills (Gibson et al., 2002; Loopman et al., 1988; Simonov and Fedotov, 1964). The land-locked lakes lose water mainly through the surface runoff in the outlet streams, and/or through evaporation over their surface. In

our study, we focus on two types of lakes, namely epiglacial and land-locked lakes, located in the ice-free area of the Larsemann Hills oasis, East Antarctic coast.

Water chemical composition and the presence of living forms in Antarctic lakes are strongly linked to their thermal regime and water balance (Castendyk et al., 2016; Bomblies et al., 2001). Among other parameters, water transport/exchange time scales are needed to study lake eutrophication, bioproduction and geochemical processes by numerical modelling

(Nuruzzama et al., 2020; Geyer et al., 2000; Foy, 1992; Burton, 1981). The lake retention time (LRT), also called "the flushing time" in Geyer et al. (2000) or "the coefficient of water external exchange" in Doganovsky and Myakisheva (2015), is among other transport scales a factor to be taken into account when modelling the water exchange and mixing processes in lakes and estuaries (Monsen et. al., 2002; Lincoln et al., 1998). It indicates the time period of water renewal in the lake (Pilotti et al., 2014; Rueda et al., 2006) and is usually expressed in years. There are only a few studies addressing

estimates of water transport scales for Antarctic lakes, mostly due to a lack of hydrological observations. For example, Foreman et al. (2004) presented hydraulic residence times (which is the same as LRT) for three lakes located in the Antarctic Dry Valleys, and Loopman and Klokov (1988) estimated the coefficient of water external exchange (which is the inverse of the LRT) for six lakes located in the Schirmacher oasis (East Antarctica).

In the Larsemann Hills oasis (East Antarctica), water temperature regime, chemical composition and biota of the lakes have

been actively studied since the 1990s (Hodgson et al., 2006 and 2005; Verleyen et al., 2004 and 2003; Saabe et al., 2004 and 2003; Kaup and Burgess, 2002; Gasparon et al., 2002; Burgess and Kaup, 1997). However, understanding of the seasonal water cycle of these lakes is still poor, due to serious gaps in the hydrological measurements in the lakes. This limits the applicability of water balance and biogeochemical models (Nuruzamma et al., 2020; Kaup, 2005). This study aims to evaluate the lake retention time of the lakes located in the Larsemann Hills oasis. We suggested to estimate the LRT from

the outflow and inflow terms of the water balance equation depending on a type of lake (epiglacial and land-locked). Our study focuses on the lakes Stepped, Nella/Scandrett, Progress, Sarah Tarn and Reid, since their water resources and biogeochemistry are important for human activity (Sokratova, 2011; Burgess and Kaup, 1997; Burgess et al., 1992).

Hydrological data collected during four summer seasons in the years 2011–2017 were used to calculate the LRT. This study is the first estimations of the LRT for the lakes located in the Larsemann Hills.

## 1. Study area

The Larsemann Hills occupy an area of approximately 50 square kilometres on the sea shore of Princess Elizabeth Land, East Antarctica. The area consists of the Stornes, Broknes and Mirror peninsulas, together with a number of small islands in Prydz Bay. The peninsulas are rocks exposed by glacial retreat since the Last Glacial Maximum (Hodgson et al., 2005). The basement geology consists of a composite of orthogneisses overlying various pegmatites and granites (Geological map, 2018; Carson et al., 1995).

The climate of the Larsemann Hills is influenced by katabatic winds blowing from the north-east during most of the austral summer. During this period, daytime air temperatures frequently exceed 10 ℃, with a mean monthly temperature of about 0 ℃. Mean monthly winter temperatures range between –15 ℃ and –18 ℃. The annual precipitation is 159 mm (statistics taken from the Russian Arctic and Antarctic Research Institute, http://www.aari.aq). Rain is rarely observed over the Antarctic ice-free areas, also known as oases. There are two meteorological stations in the Larsemann Hills. Zhongshan station (WMO index 89573) started operating in 1989, and the Progress station (89574) operated intermittently from 1988 to 1998 and since 1999 has provided continuous observations (Turner and Pendlebury, 2004). Local climatology for the period 1988–2010 was reported by the Russian Arctic and Antarctic Research Institute from the Progress station, see Table 1 in Shevnina and Kourzeneva (2017). Yu et al. (2018) reported increasing trends in annual precipitation for the period 2003–2016.

There are more than 150 lakes in the Larsemann Hills oasis. Many of them occupy local depressions formed after melting of the continental ice sheet (Gillieson et al., 1990). The lake water chemical composition is affected by sea sprays, local geology and periodic seawater surges caused by calving of the Dålk glacier situated next to the eastern corner of the oasis (Kaup and Burgess, 2002; Stüwe et al., 1989). The majority of lakes are monomictic, which means that they are thermally homogeneous during summer, also due to the persistent katabatic winds (Bian et al., 1994). The land-locked and epiglacial lakes are typical for coastal ice-free areas such as the Larsemann Hills oasis. The land-locked lakes are usually small ponds (Lake Stepped, Lake Sarah Tarn and Lake Reid in Fig. 1).

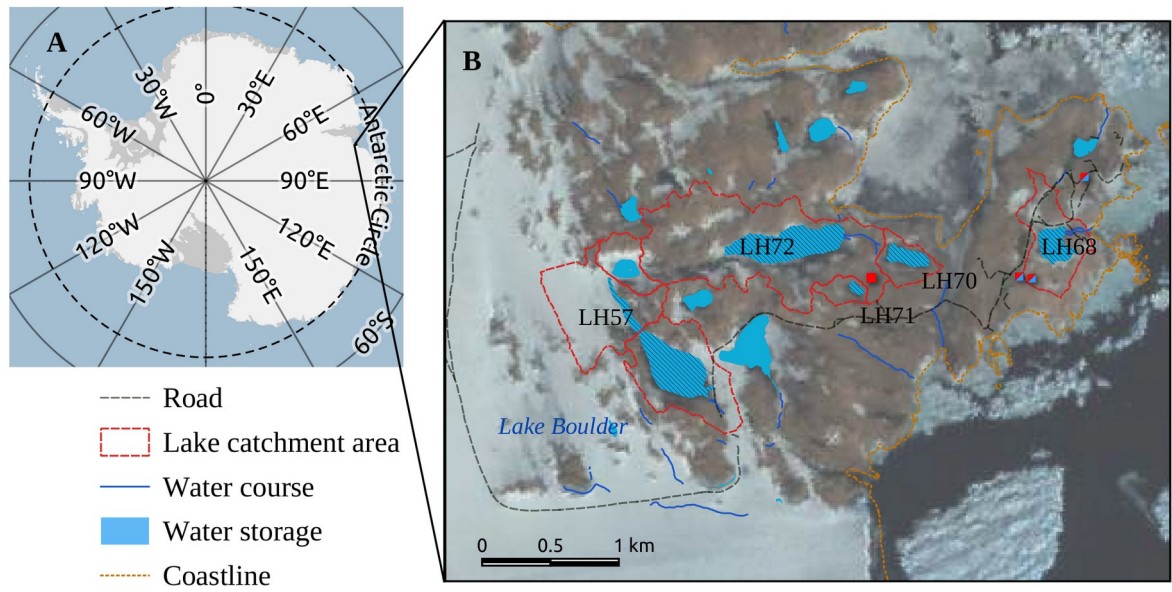

**Figure 1:** Lakes on the Mirror peninsula, the Larsemann Hills, East Antarctica:

A: the red box indicates the location of the oasis; B: the red lines outlining the catchment areas represent the lakes considered in this study according to the digital map scale of 1:25000, AAD, 2005; the LIMA composite is on the background map (Bindschadler et al., 2008). Abbreviation of a lake designation in B: LH57 – Lake Progress, LH68 – Lake Stepped, LH70 – Lake Reid, LH71 – Lake Sarah Tarn, LH72 – Lake Nella/Scandrett.

Land-locked lakes are free of ice in December–February, and their water temperatures reach 10.0–11.0 °C in January (Boronina et al., 2019; Gillieson et al., 1990). Lake Stepped is located in the old coastal lagoon, and is connected to the sea via underground leakage of water. During the warm period, an outlet stream in the north-eastern corner of the lake releases water surplus. Water surplus occurs due to melting of the seasonal snow cover in the lake catchment. Lake Sarah Tarn and Lake Reid are endorheic ponds with no (or only minor) outflow of surface/ground runoff. The land-locked lakes are fed by seasonal snow cover melting, precipitation and sea sprays (Kaup and Burgess, 2002). The water level of the lakes Stepped, Sarah Tarn and Reid varies by 0.2–0.4 m during the austral summers. According to observations and modelling results, these lakes are well mixed during the summer seasons (Shevnina and Kourzeneva, 2017). A special case is Lake Reid, which has brackish water. In this lake, thermal stratification resistant to katabatic winds of over 14 ms$^{-1}$ was observed by Kaup and Burgess (2003) in January 1994. The assumption of a mixed state is important for our calculations of LRT. However, we do not exclude Lake Reid from our calculations, and we assume it to be mixed, because this was shown by modelling results in Shevnina and Kourzeneva, (2017), although using a fresh water lake model.

The epiglacial lakes are situated at the boundary between the glaciated and rock areas, and they are mostly fed by melting of the ice/snow of the lowest (ablation) zone of the glaciers (Klokov, 1979). Lake Progress and Lake Nella/Scandrett are the

two biggest epiglacial lakes located on the Mirror peninsula of the Larsemann Hills oasis. Both lakes are fed by the melting of the Dålk glacier, although the portions of the glaciated areas differ for these lakes: it is more than 50% for Lake Progress (Fig. 1) but less than 10 % in case of Lake Nella/Scadrett. Table 1 shows the available estimates for the actual volumes of the studied lakes. Here and in the following, the lake name is given together with the lake index according to the Atlas of Lakes (Gillieson et al., 1990).

Table 1. The volume ($V$, x$10^3$ m$^3$) and area ($A$, x$10^3$ m$^2$), of five lakes located in the Larsemann Hills oasis: estimated according to (Shevnina and Kourzeneva, 2017) / (Pryakhina et al., 2020). No estimates are indicated by "–" .

| Parameter | Epiglacial lakes | | Land-locked lakes | | |
|---|---|---|---|---|---|
| | Lake Progress /LH57 | Lake Nella/Scandrett /LH72 | Lake Stepped /LH68 | Lake Reid /LH70 | Lake Sarah Tarn / LH71 |
| $V$, x$10^3$ m$^3$ | 1812.4 / 1526.75 | 1033.2 / 1490.7 | 40.5 / 51.03 | 25.5 / 40.45 | 10.5 / – |
| $A$, x$10^3$ m$^2$ | 160.6 / 125.7 | 155.9 / 157.9 | 47.3 / 44.4 | 33.1 / 35.5 | 6.1 / – |

The most recent estimates of the actual volume of the lakes Stepped/LH68, Nella/Scandrett/LH72 and Reid/LH70 are those presented by Pryakhina et al. (2020). The authors used bathymetric surveys performed on these lakes in two seasons of 2017–2018 and 2018–2019, also reported by Boronina et al., (2019). These estimates of volume are higher by 44–50 % than those given in Shevnina and Kourzeneva (2017), which are in turn partly based on the surveys of 2011–2012 made by Fedorova et al., (2012). In our study we use the estimates provided both by Shevnina and Kourzeneva, (2017) and Boronina et al., (2019), depending on the lake studied (see Methods and Data)

## 2. Methods and data

### 2.1 Data

Our estimations of the LRT were obtained from the inflow and outflow terms of a lake water balance equation. The outflow terms are evaporation and surface outflow runoff, and the inflow terms are water inflow due to melting of seasonal snow cover and precipitation over the lake surface. The components were calculated from the hydrological observations on the lakes and streams collected during four field campaigns carried out in the Larsemann Hills area.

Two campaigns lasting from 25.12.2012 to 28.02.2013 and from 03.01.2014 to 10.03.2014 were focused on measuring the lake water level/stage and water surface temperature, as well as water discharges and levels at the inlet/outlet streams, and on collecting water samples (Naumov, 2014; Vershinin and Shevnina, 2013). These observations were used to estimate the outflow terms, namely surface runoff and evaporation, for lakes in the Larsemann Hills oasis (Shevnina and Kourzeneva, 2017). In this study, we used results from (Shevnina and Kourzeneva, 2017) to evaluate the LRT from the outflow terms of the water balance equation.

Two campaigns lasted from 27.12.2011 to 05.02.2012 and from 05.01.2017 to 20.02.2017. They were mostly focused on the study of the water chemical composition in local lakes, but the snow properties were also measured, by means of snow surveys at the watersheds of selected lakes (Dvornikov and Evdokimov, 2017; Fedorova et al., 2012). During each

135 campaign, two consecutive snow surveys were performed: on 12 January and 5 February during the year 2012, and on 8–10 January and 31 January – 1 February during 2017. At the time of the first snow surveys during both campaigns, snow had partly melted in the Larsemann Hills oasis. The snow cover consisted of several stand-alone snow packs (snow fields). These snow packs contain deep, dense snow. They are persistent from year to year and probably belong to the permanent (multi-annual) cryosphere. Surveys were organized to measure the properties of the largest snow packs: their area, snow density and

140 snow depth. In 2011–2012, snow surveys were carried out on the catchment of Lake Stepped/LH68 (2 biggest snow packs). In 2016–2017, the snow surveys were performed in the catchments of Lake Stepped/LH68 (9 snow packs), Lake Reid/LH70 (7 snow packs) and Lake Sarah Tarn/LH71 (one snow pack). A map of the snow surveys is displayed in Fig. 2. In these series of snow surveys, the snow depth was measured by a metal probe with an accuracy of 0.01 m; the snow density was measured with a snow tube VS–43 (Slabovich, 1985), and the snow edge was delineated using a Garmin Etrex 30 hand-held

GPS/Glonass receiver.

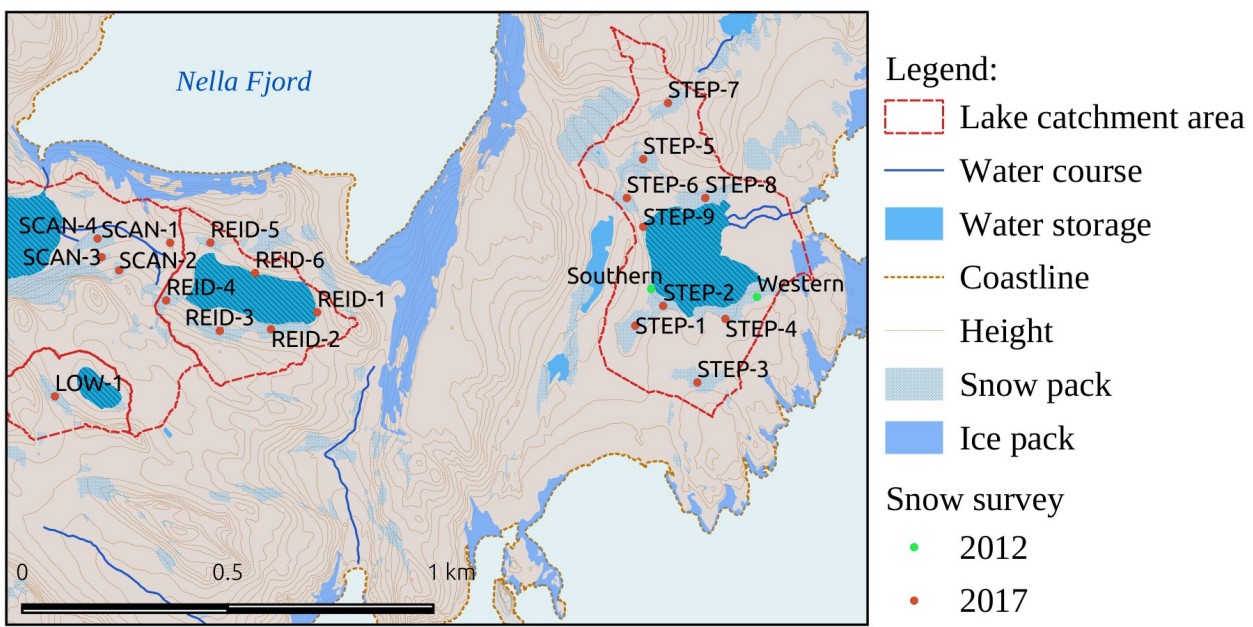

**Figure 2.** Snow surveys on the lake catchments during the campaigns of 2011-2012 (green dots) and 2016 – 2017 (orange dots). Stand-alone snow packs are identified by names for the campaign of 2011-2012 and by ID numbers for the campaign of 2017 (e.g., STEP-1 means the first snow pack in the catchment of Lake Stepped/LH68). The red line outlines the catchments, and the background map is given

according to AAD (2005).

The snow depth and density were used to calculate the snow water equivalent (SWE) accumulated in the stand-alone snow packs. An example of the SWE map obtained from the snow survey of 08–10.01.2017 in the catchment of Lake Stepped (LH68) is given in Fig. A1 in the Annex. Then, the volume of water incoming from the lake watersheds due to melting of seasonal snow cover was calculated as the SWE difference between two consecutive snow surveys. Using these SWE differences may ignores water sublimation of snow cover between the two surveys, it will lead to over- estimation of the LRT for the lakes. This water incoming is the volume of melted water which comes to the lakes from the permanent snowfields during the period between two surveys. Table A1 in the Annex provides an example of these calculations for the campaigns of 2011–2012 and 2016–2017 for Lake Stepped (LH68). This table also shows the areal retreat of the stand-alone snow packs. Details and the full data-sets from two field campaigns can be found in the reports by Dvornikov and Evdokimov (2017) and Fedorova et al. (2012). In this study, we neglected to the water sublimation over the ice covered parts of the epiglacial lakes and snow packs. It is assumed that, in summer it is small compare to others components of the lake water balance equation. However, to proof the assumption would need to a separate study. Not accounting of the water sublimation may lead to slight over- estimation of the LRT, especially for the land-locked lakes.

## 2.2 Methods

Imboden (1974) proposed the LRT as a ratio between the volume of lake epilimnion and the outflow water flux, to estimate the time scale of water renewing in lakes. Znamensky (1981) suggested the inverse of the LTR, known as "the coefficient of external water exchange", to characterize water renewal time in the lakes and reservoirs. For the lakes with thermal homogeneity, the LRT can be calculated using the lake total volume instead of the volume of lake epilimnion (Pilotti et al., 2014; Rossi et al., 1975). Both the lake volume and outflow water flux vary in time, depending on atmospheric conditions (climate and weather), topography changes, human activities and other factors. Generally speaking, this means that the LRT is also time-dependent. However, the idea of the time scale suggests that averaging over some time period should be applied. We consider the equation for the LRT for monomictic lakes as follows:

$$LRT = \frac{\bar{V}}{\bar{O}} \ , \tag{1}$$

where $LRT$ is the lake retention time, (s), $\bar{V}$ is the lake volume (m³), $\bar{O}$ is the outflow water flux (m³ s⁻¹),

$\bar{f} = \frac{1}{T} \int_0^T f(t)\,dt$ is the averaging operator, $f = \begin{bmatrix} V \\ O \end{bmatrix}$, $t$ is time, (s), and $T$ is the averaging time period, (s).

The averaging time period should be long enough. It should be longer than the LRT itself, known from its typical value or preliminary studies. For large lakes, the typical LRT value is several (tens of) years. For example, Pilotti et al., (2014) estimated LRT from the long-term average of the water discharge calculated from annual discharges. However long-term series of hydrological observations for lakes are often unavailable. If shorter time periods are considered, the result is less

reliable and more time-dependent. An example of LRT estimations from series of seasonal, monthly or daily outlet stream water discharges can be found in (Andradóttir et al., 2012). Generally speaking, for lakes with a well pronounced annual hydrological cycle, the shortest averaging time period should be one year.

The lake volume can be estimated from the height–area–volume curve depending on the lake water level/stage (Strahler, 1952). To estimate the outflow water flux, the outflow surface runoff flux is normally used (Pilotti et al., 2014, Andradóttir et al., 2012), and also evaporation (Quinn, 1992). If no long-term changes in the lake volume can be suggested, the outflow water flux in the LRT estimations can be replaced by the inflow water flux, as is done e.g. in (Doganovskiy and Myakisheva, 2015). Let us consider this in detail.

The lake water balance equation describes an evolution of the lake volume:

$$\frac{dV(t)}{dt} = Q_{in}(t) - Q_{out}(t) + \left(P(t) - E(t)\right)A(t) \pm \dots \; ,$$

where $Q_{in}$ and $Q_{out}$ are the surface inflow and outflow runoff (m$^3$ s$^{-1}$), $P$ and $E$ are precipitation and evaporation (m s$^{-1}$) over the lake surface area $A$ (m$^2$). This equation is usually applied in the integrated form, with an integration time period depending on the application, e.g. a one-year time period:

$$\frac{\Delta V}{\Delta t} = Q_{in} - Q_{out} + \left(P - E\right)A \pm D \; , \tag{2}$$

where now $\Delta V$ (m$^3$) is the volume change during the time period $\Delta t$ (e.g. 1 year), $Q_{in}$ and $Q_{out}$ are the inflow and outflow surface runoff per this time period (e.g., annual surface inflow and outflow runoff), $P$ and $E$ are precipitation and evaporation (sublimation) during this time period (e.g. annual precipitation and evaporation); and $D$ is discrepancies. The equation may be continued by adding more inflow/outflow (or accumulation/dissipation) terms depending on the lake type, the specific study case and the time scale of processes considered (Chebotarev, 1975). Contributions of some components to the total lake volume change may be minor, whereas other terms may be essential, depending on a particular case. Therefore, additional components may need to be included into the water balance equation, *i.e.* artificial water withdrawal, water sublimation over ice covered surface of a lake, underground inflow/outflow runoff, water inflow/outflow due to melting/freezing processes, etc. In our case, we apply this equation differently for the epiglacial lakes and for the land-locked lakes. For both lake types, we do not consider the underground water runoff, because most of the lakes are located on rock-covered catchments. The special case is Lake Stepped/LH68, situated in a sandy valley closed to the sea. Including no underground inflow/outflow runoff for this lake will affect the results; this is discussed in Section 3.2.

For the land-locked lakes considered in our study, the annual volume changes do not exceed 1–2 % (Boronina et al. 2019; Dvornikov and Evdokimov, 2017; Shevnina and Kourzeneva, 2017). Not all the land-locked lakes in our study have inlet streams. Therefore, the surface inflow due to melting of snow cover over the catchment area is the most important inflow

term for their water balance (Shevnina and Kourzeneva, 2017). Then, Eq. (2) for the land-locked lakes may be written as follows:

$$D = M - Q_{out} + (P - E)A \,, \tag{2a}$$

where $D$, (m$^3$) is the discrepancies including unknown outflow terms, i.e. water withdrawal ($D_{out}$) and inflow terms ($D_{in}$); $M$, (m$^3$) is the amount of water due to snow melting for this time period (e.g. per year). Then, the inflow and outflow terms in Eq. (2) can be separated:

$$D_{out} + Q_{out} + E \cdot A = D_{in} + M + P \cdot A \tag{3}$$

This allows calculation of the LTR both from the outflow or dissipation ($LRT^o$) and inflow or accumulation ($LRT^i$) terms of the water balance for the land-locked lakes:

$$LRT^o = \frac{\bar{V}}{Q_{out} + D_{out} + E \cdot A} \tag{4}$$

and

$$LRT^i = \frac{\bar{V}}{M + D_{in} + P \cdot A} \,. \tag{5}$$

Now $\bar{V}$, (m$^3$) is the mean lake volume during the considered time period (1 year). Lake Stepped/LH68 provides water supply for the Russian polar base Progress, and thus Eq. (2) needs to be extended with the water withdrawal added to the outflow terms of the water balance equation for this lake. In this study this term was not taken into account due to lack of data, and we consider this as a discrepancy.

Lakes of epiglacial type are located on the edge of ice sheets and have a specific water balance due to that fact. Their volume can change considerably during the year. Epiglacial lakes may be connected to supraglacial lakes through the ephemeral hydrological network. Abrupt drops of water level/stage are reported to occur in many epiglacial lakes located in the Schirmacher, Thala Hills and Bunger Hills oases (Gibson et al., 2002; Klokov, 1979). These drops lead to the release of a huge amount of water into the sea, in events known as water outbursts (Schomacker and Benediktsson, 2017; Klokov, 1979). In the Larsemann Hills oasis, the water drops of the level/height were reported for Lake Progress/LH57 in the years 2013 and 2014, and resulted in an outflow of 7.5–8.1 % of the total lake volume (Shevnina and Kourzeneva, 2017). An abrupt drop of the water level/stage was also reported for Lake Nella/Scandrett/LH72 in 2018, with release of 5.7 % of the lake total volume (Boronina et al., 2019). In this study, the water outburst due to the abrupt drops should be accounted for in Eq. (2) as the outflow term of the water balance equation for the epiglacial lakes. At the same time, the resulting change in the water volume between the end $t_2$ and the beginning $t_1$ of the time period (annual change) $\Delta V = V(t_2) - V(t_1)$ is minor. Then the Eq. (2) for the epiglacial lakes can be written:

$$D = Q_{out} + Q_{abr} + E \cdot A \,, \tag{2b}$$

where $Q_{abr}$, (m$^3$) is the surface outflow runoff due to the abrupt drop. This term can be calculated from the lake volume (water levels/stage) observations on dates before and after the drop, $\Delta V_{abr}$ :

$$Q_{abr} = \frac{\Delta V_{abr}}{\Delta t} \qquad (6)$$

For the epiglacial lakes, in our study the LRT was estimated only from the outflow terms of the water balance equation Eq. (2b):


$$LRT^o = \frac{\bar{V}}{Q_{out} + Q_{abr} + D + E \cdot A} \; . \qquad (7)$$

The observations from the measurement campaigns of 2012–2013 and 2013–2014 were used to calculate the $LRT^i$ and $LRT^o$ for the land-locked and epiglacial lakes by applying Eqs. (4, 5 and 7). For the evaporation term, we used the estimates given in (Shevnina and Kourzeneva, 2017). These estimates were obtained with different methods: from the empirical equation of (Odrova, 1979) and from the modelling results obtained with the lake model FLake (Mironov et al., 2005).

Data from the measurement campaigns of 2011–2012 and 2016–2017 described in Dvornikov and Evdokimov (2017), and from Fedorova et al. (2012), were used to calculate the $LRT^i$ for the land-locked lakes by applying Eq. (5). From these observations, the annual inflow due to melting of snow cover was evaluated. In addition, the annual precipitation amount was calculated using the observations on the nearest meteorological site Progress, located about 700 m from lake Stepped/LH68. The monthly sum of the precipitation observed at this site is available at www.aari.aq/data/progress/prec.txt.

We used the precipitation sum for three summer months in Eq. (5) because the lakes are covered by ice during the rest of the year. The precipitation during this period is usually blown away from the lake surface.

Table 2 summarizes methods used in our study to estimate the LRT for various lakes together with the years of the measurement campaigns which provided the data, and also presents some data. We used the estimates of lake volume/area given by Boronina et al. (2019) for Lake Stepped/LH68 and Lake Reid/LH70, because they use measurements performed in

the next season after the last measurement campaign. The volume/area data of Lake Nella/Scandrett/LH72 were also taken from Boronina et al. (2019) because they used a bathymetric survey, which promises good accuracy. For the rest of the lakes, the volume/area estimates were taken from Shevnina and Kourzeneva (2017).

Table 2. The volume ($V$, x10$^3$ m$^3$) and area ($A$, x10$^3$ m$^2$) and the methods used to evaluate the lake retention time (LRT, years) of five lakes located in the Larsemann Hills oasis.

| Parameter | Epiglacial lakes | | Land-locked lakes | | |
|---|---|---|---|---|---|
| | Lake Progress / LH57 | Lake Nella/Scandrett /LH72 | Lake Stepped /LH68 | Lake Reid /LH70 | Lake Sarah Tarn / LH71 |
| $V$, x10$^3$ m$^3$ | 1812.4 | 1490.7 | 51.03 | 40.45 | 10.5 |
| $A$, x10$^3$ m$^2$ | 160.6 | 157.9 | 44.4 | 35.5 | 6.1 |

| | | | | | |
|---|---|---|---|---|---|
| Equation for $LRT^o$ / $LRT^i$ | (7)/- | (7)/- | (4)/(5) | (4)/(5) | (4)/(5) |
| Years of campaign for $LRT^o$ | 2012-2013 2013-2014 | 2012-2013 2013-2014 | 2012-2013 2013-2014 | 2012-2013 2013-2014 | 2012-2013 2013-2014 |
| Years of campaign for $LRT^i$ | – | – | 2011–2012 2016–2017 | 2016–2017 | 2016–2017 |

Shevnina and Kourzeneva (2017) provided some details of errors and discrepancies of the water balance components. In our study, the discrepancies (*D*) were not taken into account in the estimates of the LRT. This leads to over- estimation of the LRT in our calculations, since the discrepancies of the water balance equation are always in the denominator. The polar environment limits the water exchange in local lakes to the warm season, usually coinciding with the summer season (December–February/June–August in the Southern/Northern Hemisphere). In our calculations we used the seasonal

estimates of the outflow and inflow terms, equating them with the whole year because we assumed no water flux for Antarctic lakes during the frozen period. Even it may be a substantial (Kaup and Haendel, 1995), the evaporation during the frozen period (water sublimation) was also not accounted because no observations were available for its evaluations. The hydrological observations during field campaigns covered almost, but not exactly, the whole warm season. They lasted 38– 65 days in December–March, depending on the available logistics for a field campaign. The length difference between the

observation period and the warm season usually leads to an over-estimation of the LRT in the case of land-locked lakes. Furthermore, not accounting for the water withdrawal from Lake Stepped/LH68 leads to an over- estimation of the $LRT^o$ for this lake.

## 3. Results

### 3.1 Retention time of the epiglacial lakes

The epiglacial lakes Progress/LH57 and Nella/Scandrett/LH72 are among the biggest water bodies of the Larsemann Hills oasis (Gillieson et al., 1990). Their seasonal water cycle starts in December with water accumulation due to melting of the seasonal snow/ice cover in the lake watersheds. In the first week of January, the volume of these lakes reaches a maximum and then rapidly decreases due to water outburst into the sea (Boronina et al., 2019; Shevnina and Kourzeneva, 2017),

leading to an abrupt drop of the water level/stage in these lakes. The seasonal cycle ends in mid-February with freezing of the lakes. During the warm season, the surface area of these lakes is up to 60–80 % free of ice.

The main contribution to the volume changes of these epiglacial lakes during the warm season is the outflow runoff, and the role of the evaporation over the lake surface is considered to be minor. Table 3 shows the $LRT^o$ of the epiglacial lakes for the years 2012–2013 and 2013–2014, together with the measured outflow surface runoff and evaporation from the lake surface calculated by two methods: according to (Odrova, 1979) and using the modelling results of the lake model FLake (Mironov et al., 2005). All the values in these calculations are taken from (Shevnina and Kourzeneva, 2017): the surface outflow runoff $Q_{out}$, runoff due to the abrupt drop $Q_{abr}$, and the evaporation $E \cdot A$ calculated by two methods. Note that $Q_{out}$ and $E \cdot A$ provided in (Shevnina and Kourzeneva, 2017) and in Table 1 are integrated for the period of the measurement campaign/warm season, although we used them to characterize the whole year (see explanations in Section 2.2). The estimated $LRT^o$ of the biggest Lake Progress/LH57 is 12–13 years. For the second biggest Lake Nella/Scandrett/LH72, the $LRT^o$ was estimated as 6–7 years. The estimations are very similar for data from both campaigns.

Table 3. The $LRT^o$ (years) of two epiglacial lakes, calculated from the measurement campaign data of 2012–2013 and 2013–2014, the measured outflow surface runoff $Q_{out} + Q_{abr}$ (x10³ m³ per season) and evaporation from the lake surface $E \cdot A$ in the volumetric rate (x10³ m³ per season) calculated according to (Odrova, 1979) / (Mironov et al., 2005). No observations are indicated by "–" .

| Lake | Year of campaign | $Q_{out} + Q_{abr}$ | $E \cdot A$ | $Q_{out} + Q_{abr} + E \cdot A$ | $LRT^o$ |
|---|---|---|---|---|---|
| Nella/Scandrett/LH72 | 2012–2013 | 246 | – | ~246 | 6 |
| | 2013–2014 | 190 | 7.6 / 18.3 | 198 / 209 | 7 / 7 |
| Progress/LH57 | 2012–2013 | 150 | – | ~150 | 12 |
| | 2013–2014 | 128 | 7.7 / 18.8 | 136 / 147 | 13 / 12 |

The errors of the LRT estimates depend on the errors of both hydrological/meteorological measurements and calculation methods. Here, the $LRT^o$ of two epiglacial lakes was estimated from the hydrological measurements collected by various methods. The programme of hydrological observations varies from season to season. The field campaigns of 2012–2013 and 2013–2014 covered only 60–66 days in the austral summer. Therefore, some over- estimation may be expected due to lack of data in the start and end of the warm season. In both seasons, the water exchange cycles in the epiglacial lakes started already in December, and were thus longer than the periods covered by the observations. The hydrological

observations were made with the same methods and tools in both seasons: the daily outflow water discharges were calculated from the water discharges and water level/stages measured in the outlet streams (Guidelines, 1978). The water discharge in the stream was calculated from the flow velocity measured at one chosen point, and its maximum error was estimated as *ca.* 10 % (Zhelezhnjakov and Danilevich, 1966). The actual volumes of these lakes were estimated from the bathymetric surveys in 2011 and 2018 with similar measurement tools and the data-processing technique 3D Analyst by ESRI (https://www.esri.com). We can expect 10% accuracy for the volume estimations. This actually leads to a 10% accuracy for the estimates of the $LRT^o$ in the case of the epiglacial lakes. To further improve the estimates of the $LRT^o$, long-term hydrological observations covering the whole warm season are needed. Observations during the campaign of 2012–2013 did not allow calculation of evaporation over the lake surface. The $LRT^o$ was estimated without taking evaporation into account, which leads to some over-estimation. However, for the season 2013–2014, different methods to calculate the evaporation, namely according to the empirical formula by (Odrova, 1979) and from the modelling results with the lake model FLake (Mironov et al., 2005), did not have a great effect on the $LRT^o$ estimates for epiglacial lakes, resulting in a difference of only 1 year for Lake Progress/LH57. This is because the outflow surface runoff from these lakes is much greater than the evaporation, and uncertainties in the evaporation calculation do not much affect the accuracy of the $LRT^o$ estimations.

### 3.2 Retention time of the land-locked lakes

We estimated the $LRT^o$ using Eq. (4) for three land-locked lakes: Lake Stepped/LH68, Lake Reid/LH70 and Lake Sarah Tarn/LH71. Lake Stepped/LH68 is a source for the stream flowing during almost 3–4 months in the austral summer. Its water discharge varies with the water level/stage. Lakes Reid/LH70 and Sarah Tarn/LH71 are endorheic ponds, *i.e.* with only minor contribution of the outflow surface/underground runoff to the changes in lake volume. Therefore, the $LRT^o$ was estimated with only seasonal evaporation, which was considered to be the main loss term of the water balance equation for these lakes. The hydrological observations collected during two field campaigns for 2012–2013 and for 2014 were used to calculate the $LRT^o$. Observations over Lakes Reid/LH70 and Sarah Tarn/LH71 were performed only during the field campaign of 2014.

All the values in the calculations were taken from (Shevnina and Kourzeneva, 2017): the lake mean volume $\bar{V}$, the surface outflow stream runoff $Q_{out}$ and the evaporation $E \cdot A$. The errors inherent to the estimations from the uncertainty in the evaporation calculation were studied by the use of different methods to evaluate the evaporation: according to the empirical equations of (Odrova, 1979) and from the modelling results of the lake model FLake (Mironov et al., 2005). Shevnina and Kourzeneva (2017) presented more details of the methods applied. Table 4 shows the $LRT^o$ of the land-locked lakes

calculated using Eq. (4) from the measurement campaign data of 2012–2013 and 2014, together with the measured outflow surface runoff and evaporation from the lake surface calculated by two methods. All the outflow terms of the water balance equation provided in (Shevnina and Kourzeneva, 2017) and in Table 4 are integrated for the period of the measurement campaign/warm season, although we used them to characterize the whole year period (see explanations in Section 2.2).

Table 4. The $LRT^o$ (years) of three land-locked lakes, calculated from the measurement campaign data of 2012–2013 and 2014, the measured outflow surface runoff $Q_{out}$ (x10$^3$ m$^3$ per season) and evaporation from the lake surface $E \cdot A$ in the volumetric rate (x10$^3$ m$^3$ per season) calculated according to (Odrova, 1979) / (Mironov et al., 2005).

| Lake | Year of campaign | $Q_{out}$ | $E \cdot A$ | $Q_{out} + E \cdot A$ | $LRT^o$ |
|---|---|---|---|---|---|
| Stepped/LH68 | 2012–2013 | 26.6 | 5.5 | 32.1 | 1.6 |
| | 2013–2014 | 20.7 | 5.7 / 5.5 | 26.4 / 26.2 | 2 / 1.6 |
| Reid/LH70 | 2013–2014 | 0 | 4.6 / 6.4 | 4.6 / 6.4 | 9 / 4 |
| Sarah Tarn/LH71 | 2013–2014 | 0 | 0.5 / 0.7 | 0.5 / 0.7 | 21 / 15 |

In this study, the land-locked Lake Reid/LH70 and Lake Sarah Tarn/LH71 were considered as endorheic ponds, and therefore only evaporation was accounted for in the $LRT^o$ as an outflow term. We estimated the $LRT^o$ for Lake Sarah Tarn/LH71 as 15–21 years, and for Lake Reid/LH70 as 4–9 years, depending on the method applied to calculate the evaporation: according to the simulations with the lake model FLake or with the empirical equation. The estimated $LRT^o$ is sensitive to the method of calculating evaporation for these two endorheic lakes: the estimated $LRT^o$ values differed by a factor of *ca.* 1.5-fold when the different evaporation calculation methods were used. The method using the model FLake (Mironov et al., 2005) gave a shorter time period of water exchange for these lakes compared to Odrova (1979).

The difference of the $LRT^o$ estimates for the land-locked lakes gives an idea about the errors. There are no direct measurements of evaporation for these lakes, and therefore it is difficult to quantify the errors and uncertainties for evaporation calculation for each method. Tanny et al. (2011) suggested that the difference between the directly measured daily evaporation figures and those calculated from the semi-empirical equations varies from 15 % to 45 %, and the energy budget method generally gives better accuracy. Therefore, we decided to use the results obtained with the lake model FLake (Mironov et al., 2005) when estimating the $LRT^o$ for the land-locked lakes.

The lake retention time estimated from the water balance inflow (or accumulation) terms was calculated using Eq. (5). For this estimation, the hydrological observations collected during the field campaigns in 2011–2012 and 2016–2017 were used.

The amount of water incoming due to melting of snow cover $M$ was calculated from the snow surveys in the watersheds of Lake Stepped/LH68, Lake Reid/LH70 and Lake Sarah Tarn/LH71 (Dvornikov and Evdokimov, 2017; Fedorova et al., 2012). During the campaign of 2011–2012, observations were performed only over the watershed of Lake Stepped/LH68, whereas during 2017 measurements were made over the watersheds of all three lakes. Since melting occurs only during the spring period, this value characterizes the annual volume of water incoming due to the melting of snow cover. Another inflow term in Eq. (5) is precipitation over the lake surface, $P \cdot A$. We considered only precipitation during the warm season, when lakes are free of ice, because during the cold season the solid precipitation is mostly blown away from the ice-covered lake surface by katabatic winds. In the season 2016–2017, overall precipitation was calculated as the sum of the monthly precipitation observed in December, 2016, and January – February, 2017 at the meteorological site Progress. In the season 2011–2012, the monthly precipitation was not reported at this site, and thus we used the average monthly values calculated for the period of 2003–2017. The mean volume $\bar{V}$ and area $A$ of the considered lakes were taken from (Shevnina and Kourzeneva, 2017). Table 5 shows the $LRT^i$ of the land-locked lakes estimated from the data of two field campaigns, together with accumulation terms: the inflow surface runoff due to melting of snow cover $M$ and annual precipitation over the lake surface. All the accumulation terms in Table 5 reflect the period of the measurement campaign/warm season, although we used them to characterize the whole annual period (see explanations in Section 2.2).

Table 5. The $LRT^i$ (years) of three land-locked lakes, calculated from the measurement campaign data of 2011–2012 and 2016–2017, the inflow runoff due to melting of snow cover estimated from the snow surveys $M$ (x10$^3$ m$^3$ per season) and precipitation over the lake surface $P \cdot A$ in the volumetric rate (x10$^3$ m$^3$ per season).

| Lake | Year of campaign | $M$ | $P \cdot A$ | $M + P \cdot A$ | $LRT^i$ |
|---|---|---|---|---|---|
| Stepped/LH68 | 2011–2012 | 0.55 | 1.18 | 1.73 | 30 |
|  | 2016–2017 | 2.56 | 1.32 | 3.88 | 13 |
| Reid/LH70 | 2016–2017 | 1.02 | 0.92 | 1.95 | 21 |
| Sarah Tarn/LH71 | 2016–2017 | 0.14 | 0.17 | 0.31 | 34 |

Tables 4 and 5 show a considerable difference in the LRT calculated from the outflow and inflow terms of the water balance equation for the land-locked lakes. In the case of Lake Stepped/LH68, the estimated $LRT^o$ is less than 2 years for both measurement campaigns and both methods to calculate evaporation, whereas the estimated $LRT^i$ is 13–30 years, depending on the measurement campaign. For lakes Reid/LH70 and Sarah Tarn/LH71, the highest estimated $LRT^o$ is *ca.*

2 times smaller than $LRT^i$. This difference is caused by the quality of the hydrological observations available for the analysis (the periods of the field campaigns and the hydrological characteristics measured in the particular campaign). The $LRT^o$ was evaluated from the hydrological measurements covering the period of 60–66 days of the austral summers of

2012–2013 and 2013–2014. The hydrological observations include the water level/stage in the stream originating from Lake Stepped/LH68 as well as the water discharges in this stream. This permitted estimation of the outflow surface runoff with an accuracy of 7–10 % (Guidelines, 1978) within the observed periods. Therefore, we suggest that the LRT calculated from the outflow terms of the water balance equation are more reliable estimates. Lake Stepped/LH68 serves as the technical water supply for the Progress station (Russia), although this outflow term was not accounted for in this study. This leads to an

over- estimation of the $LRT^o$ for this lake. In the future, measurement of the volumes of water extracted from Lake Stepped/LH68 will allow improving the estimations given in this study. Omission of the outflow underground runoff also affects the estimations of the LRT for Lake Stepped/LH68, leading to its over- estimation. To improve the estimates, specific measurements are needed.

The $LRT^i$ was evaluated from only two snow surveys in January–February 2012 and January–February 2017 with an

interval of 20–23 days, and their dates were quite far from the beginning and the end of the astral summer. By the time of the first survey, much of the snow cover in the catchments of the studied lakes had already melted (Dvornikov and Evdokimov, 2017; Naumov, 2014; Vershinin and Shevnina, 2013). Only the snow melt from the remaining (possibly permanent) snow packs was estimated during these surveys. This led to significant over- estimations of the $LRT^i$. Furthermore, the large difference between the water balance outflow terms and the inflow term, namely the surface inflow

due to melting of seasonal snow cover (*M)* estimated from the measurements over residual snow packs, means that these snow packs do not serve as a main source of water inflow to the considered lakes. We suggest that melting of seasonal snow cover over the territory of the whole watershed is of higher importance. An earlier date of the first snow survey, and a larger area of measurements and remote sensing methods will be applied in future studies. These will help to improve the LRT estimates from the inflow terms and to understand better how melting of seasonal/permanent snow contributes to the water

balance of these lakes.

## 4. Discussion

It is recognized that liquid water presence in the Antarctic ice-sheet is more extensive now than ever observed previously (Stokes et al., 2019; Bell et al., 2019). The melted water feeds a population of lakes associated with glaciers and connected by ephemeral stream flows into the hydrological network, which becomes well developed in the warmest summers

(Holgson, 2012; Klokov, 1979). The number of hydrological observations on these glacier-related lakes and streams is extremely limited, and they are restricted to the measurements collected in seasonal field campaigns. However, they provide irreplaceable data for rapid detection of the amount of liquid water melted over ice sheets (Antarctic and Greenland). The

hydrological observations include the bathymetric surveys on the lakes, the snow surveys on their catchments, the measurements of lake water level/stage, lake water temperature, and water discharges in the inlet/outlet streams.

In Antarctica, long-term hydrological observations (since the late 1980s) are available only for several lakes located in the East of the continent. In this study, the hydrological observations collected in four summer field campaigns in the Larsemann Hills oasis (East Antarctica) were used to estimate the components of the water balance equation: lake volume, surface outflow runoff, evaporation, precipitation and inflow due to melting of snow cover. Then, the water transport time scale (namely the lake retention time) was evaluated from the components separately for the epiglacial and land-locked

lakes. Among the lakes studied, the two epiglacial lakes are biggest in volume, with LRTs varying between 6 and 12 years. The LRT of the land-locked lakes is from 2 to 15 years. Long water transport times are common for the endorheic lakes since they lose water only by evaporation, which is expected be low in the cold polar environment. It should be noted that in this study we did not account for sublimation from the lake surface during the ice-covered period, although this could be a significant outflow term of the overall water balance for polar lakes (Huang et al., 2019; Faucher et al., 2019; Kaup and

Haendel, 1995). This can lead to over- estimations of the LRT calculated from the outflow terms of the water balance equation both for the epiglacial and land-locked lakes.

Until recently, little was known about the water transport time scales of the glacial lakes located in Antarctica, which makes it difficult to analyse our results by comparing them with other studies. Foreman et al. (2004) applied the longest hydrological observations for evaluating the hydraulic residence time of three large epiglacial lakes located in the Dry

Valley, East Antarctica (Lake Bonney, Lake Hoare and Lake Fryxell). The estimated hydraulic residence time for Lake Bonney is 77 years, for Lake Hoare 281 years and for Lake Fryxell 107 years. The authors observed that the estimations of the residence time depended on the series length and the period of the hydrological observations included in the analyses. In the case of Lake Fryxell, for example, the hydraulic residence time varied from 107 to 9 years when calculated using data for the years 1995–2001 and the year 2001–2002, respectively. The volumes of these lakes are much higher than those of

the lakes located in the Larsemann Hills oasis. All this makes the comparison between our estimates and those presented by Foreman et al. (2004) difficult.

The hydrological observations on six lakes and streams located in the Schirmacher oasis (East Antarctica) date back to the early 1980s. These observations covered the whole hydrological season lasting from November, 1983 to March, 1984. Further they were used by Loopman and Klokov (1988) to estimate the water exchange coefficient (the inverse of LRT) for

three epiglacial lakes: Lake Smirnova with a volume of $613 \times 10^3$ m$^3$, Lake Pomornik with a volume of $236 \times 10^3$ m$^3$, and Lake Glubokoe with a volume of $1930 \times 10^3$ m$^3$. The estimated LRTs were 1 and 2.4 years for Lake Smirnova and Lake Pomornik, respectively (Kaup, 2005). The volume of these lakes is much less than the volume of Lake Nella/Scandrett/LH72 or Lake Progress/LH57. The LRT of Lake Glubokoe, which is comparable in volume with Lake Progress/LH57, is estimated as 2.6 years, and it is almost threefold less than for Lake Progress/LH57. It is interesting to note that Lake Progress is connected

with the epiglacial Lake Boulder, which received the outburst flood in January 2018 (Popov et al., 2020). This was the first observed outburst on this lake since the start of instrumental observations in the Larseman Hills oasis in the early 1990s.

The results show a clear difference in the LRT estimated for the land-locked lakes Stepped/LH68, Sarah Tarn/LH71 and Reid/LH70. The LRT estimates depend to a great extent on the methods: for Lake Stepped/LH68, the LRT estimated from the outflow surface runoff is less than 2 years, whereas when estimated from the inflow due to melting of seasonal snow

cover it is 13–30 years. For the endorheic Lake Sarah Tarn/LH71 and Lake Reid/LH70, the LRT estimated from the outflow term is also dependent on the method used to calculate evaporation. If an empirical approach is used to calculate evaporation, the estimated LRT is shorter, being 15 and 4 years for Lake Sarah Tarn/LH71 and Lake Reid/LH70, respectively. The LRTs estimated from the inflow terms of the water balance are 34 years and 13 years, respectively, for these two lakes. We think that this difference is caused by the quality of the hydrological observations over snow. In

particular, the length of the period and the start/end dates depend on the schedule for seasonal logistics for each field campaign. We propose relying on the estimates from the outflow surface runoff, since they are based on hydrological measurements collected during longer periods (60–66 days). Data collected during the snow surveys in the seasons 2011–2012 and 2016–2017 are not sufficiently representative to estimate the inflow to the lakes due to melting of seasonal snow cover over their watersheds. Also, water loss through sublimation over the ice/snow cover in the catchment area can be

substantial (Hermichen et al., 1985).

The LRTs of three land-locked lakes varied from 2 to 15 years. The longest retention time is typical for endorheic lakes losing water via evaporation/sublimation. In this case, errors in the LRT estimates depend on the methods used to evaluate the evaporation. Hitherto, there are still only a few studies addressing evaporation over the Antarctic lakes. Most of them use the energy budget or semi-empirical equations to evaluate the evaporation from the meteorological observations (Dhote

et al., 2020; Faucher et al., 2019). Borghini et al. (2013) evaluated the evaporation over three shallow land-locked lakes located in Victoria Land, East Antarctica. The surface area of these endorheic lakes has not substantially decreased since the late 1980s. The authors suggested that water loss through surface evaporation accounts for *ca.* 40–45 % of the overall change in lake volume. In the Larsemann Hills, no significant changes in the surface area of the endorheic lakes Sarah Tarn and Reid have been observed since the late 1980s (Shevnina and Kourzeneva, 2017), and the contribution of evaporation

may be significant in the water balance of these lakes.

It should be noted that direct measurements of the evaporation over ice/snow and lakes are still rarely found for the remote polar regions, mostly due to the available measuring techniques. The traditional pan-evaporators are difficult to deploy and operate. The method of eddy covariance provides the best accurate estimates for the evaporation rate for the lakes. This method gives some insight into the surface energy balance with the unique measurements, and allows improving the semi-

empirical equations with best estimates of the regional coefficients (Tanny et al., 2008). In future works we shall study evaporation over the epiglacial and land-locked lakes with different methods: eddy covariance, energy budget and semi-empirical methods. The method of eddy covariance will provide a reference while analysing the uncertainties in evaporation

and the LRT estimates of the Antarctic lakes. Since the water sublimation over ice covered lakes depends on meteorological conditions (air temperature and humidity, wind speed) which are largely similar in the East Antarctic oases, our next study address to evaluation of water loss for ice free and ice covered lakes. We would expect that the over- estimation of the LRT due to neglected sublimation would not exceed 10 % (Shevnina et al., 2020).

**5 Conclusion**

Water chemical composition and the presence of life in Antarctic lakes are strongly connected to the lake water balance and thermal regime. The modelling approach is applied along with others in understanding eutrophication, bio production and geochemical processes in lakes. Many geochemical models need characterization of water transport/exchange time scales. This study provides the first estimates for water transport time scale characteristics, namely the lake retention time, for five lakes located in the Larsemann Hills oasis, East Antarctica. The LRT was evaluated depending on the lake type: separately for the epiglacial and the land-locked lakes. The LRT was estimated with two methods based on the inflow and outflow terms of the water balance equation. The outflow terms are evaporation and surface outflow runoff; the inflow terms are the water inflow due to melting of snow cover and summer season precipitation over the lake surface. To calculate these components, we used the hydrological observations on the lakes and streams collected during four field campaigns in the austral summers of 2011–2017.

Our study showed that the LRTs of lakes in the Larsemann Hills oasis are very different, depending on the lake type. This is because the mechanisms of water exchange differ for the epiglacial and land-locked lakes. The epiglacial lakes lose water mostly through the outflow surface runoff. For these lakes, the surface runoff is much higher than the evaporation over the lake surface area. For the endorheic land-locked lakes, the evaporation over the surface area is the main loss term of the water balance equation. The obtained LRT indicates that water exchange in epiglacial lakes is faster than in the land-locked endorheic lakes in the Larsemann Hills oasis, sometimes much faster. For example, the estimated LRT of the large epiglacial lake Nella/Scandrett/LH72 is 6–7 years, and of the small land-locked lake Sarah Tarn/LH71 15–21 years. Therefore, different strategies should be applied to monitor the hydrological cycle of these lakes in general, and to obtain better estimates of the LRT in particular. For the epiglacial lakes, accurate measurements of water discharges to inlet/outlet streams are needed, covering most of the warm season. If data are missing in the start or end of the warm period, over- estimation of the LRT may be expected. For the endorheic land-locked lakes, methods applied to measure/calculate evaporation are of great importance. All these methods need meteorological observations. To choose the best technique to calculate evaporation, measurement campaigns with eddy covariance flux observations would be very helpful. The role of the water sublimation over the ice-covered surface of the lakes should also be studied in details. It may be substantial component of a lake water balace equation integrated over the period of year. For land-locked lakes, the LRT can also be calculated from the inflow term of the water balance equation, namely from snow melting. However, snow surveys for that purpose should start early enough, before the onset of the melting season, and also end late enough. We recommend to carry out the first snow

survey already in early December and the second one at the end of February. Otherwise, we may over-estimate the LRT considerably, as was the case in this study.

   Hydrological observations in the epiglacial lakes also allow accurate evaluation of the amount of liquid water seasonally melting from the margins of glaciers. Observations of the lake water level/stage and temperature are of particular importance, in addition to water discharges in inlet/outlet streams. These measurements, complemented with in-situ

bathymetric surveys, will allow verification of remote sensing measurements of glacier melting. We would consider our LRT estimates as a preliminary attempt suggest two versions of the water balance equation depending on a lake types namely for the epiglacial and land-locked lakes. To improve these estimates, hydrological and meteorological observations are needed in the lakes and streams, lasting for the whole warm season. We recommend the inclusion of hydrological observations in future field campaigns which are planned for the Larsemann Hills oasis. The uniform observational programme for the

hydrological measurements will contribute to better estimations of the water transport time scale. We would especially propose to install the hydrological monitoring network in Lake Stepped/LH68 and Lake Progress/LH57, due to their importance for the water supply of the Progress and Zhongshan scientific stations.

**Annex**

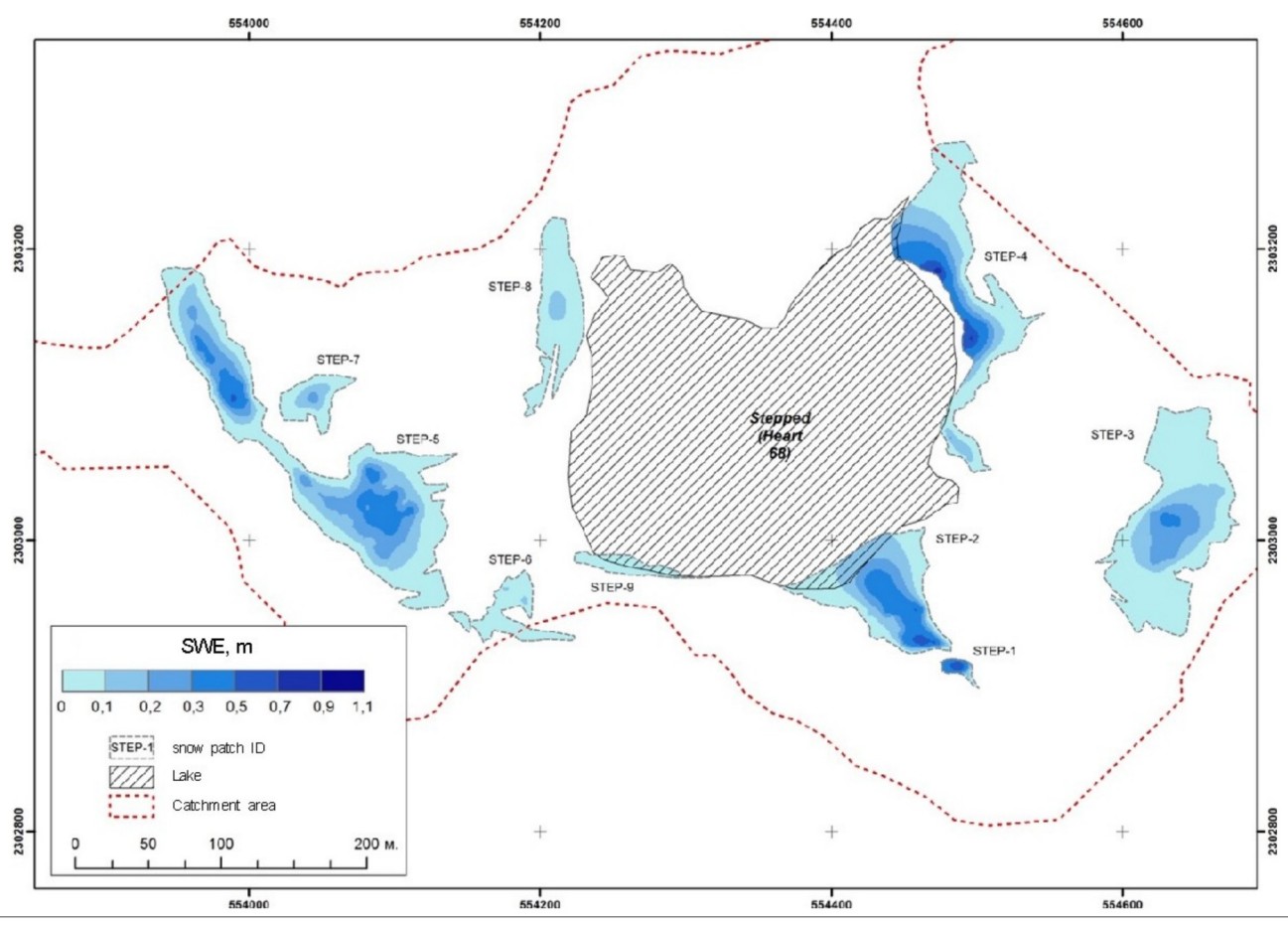


Figure A1: A map of SWE calculated from the snow surveys of 08–10.01.2017 over 9 stand-alone snow packs on the watershed of Lake Stepped, according to (Dvornikov and Evdokimov, 2017).

Table A1. The volume of water from melting of the stand-alone snow packs and incoming into Lake Stepped/LH68 together with the melted area between 2 snow surveys for each stand-alone snow pack for 2 measurement campaigns during the years 2011–2012 and 2016–2017. Snow pack indices can be obtained from Fig. 3 and Fig. A1.

| Year/campaign | Snowfield index | Melted area, m² | Volume of water melted, m³ |
|---|---|---|---|
| 2012 | Western | 2484 | 260 |
| | Southern | 6929 | 286 |
| | Total volume of water melted: | | 546 |

| | STEP-1 | 218 | 62 |
|---|---|---|---|
| 2017 | STEP-2 | 2669 | 638 |
| | STEP-3 | 2436 | 276 |
| | STEP-4 | 2157 | 438 |
| | STEP-5 | 4201 | 1020 |
| | STEP-6 | 1293 | 28 |
| | STEP-7 | 419 | 32 |
| | STEP-8 | 1971 | 64 |
| | STEP-9 | 516 | 4 |
| | Total volume of water melted: | | 2562 |

**Data Availability**

The data used in our calculations of the volume of water incoming due to melting of snow cover in January 2017 are provided as a supplement to this manuscript (Supplement_Shevnina_etal2021.xlsx).

**Authors' contributions**

The original idea of this paper was developed by Elena Shevnina. Elena Shevnina and Ekaterina Kourzeneva wrote the text, with equal contribution. Elena Shevnina performed the calculations and prepared the tables and figures. Yury Dvornikov

and Irina Fedorova contributed with estimations of the volume of water incoming due to melting of seasonal snow cover in the seasons 2011–2012 and 2017.

**Competing interest**

No potential conflict of interest is reported by the authors.

**Acknowledgements**

This study was supported by the Academy of Finland (contract number 304345), and the field campaigns in the Larsemann Hills oasis are organized according to the logistics of the Russian Antarctic Expedition. Our special thanks go to A. Evdokimov, A. Krasnov and A, Zubov, who carried out the snow surveys in 2012 and 2017. We also thank the participants of EGU2019 (May, 2019, Vienna, Austria) and 6th workshop on "Parameterization of Lakes in Numerical Weather

Prediction and Climate Modelling" (October, 2019, Toulouse, France) for their questions and discussion. We thank Martin

Truffer and Enn Kaup for their comments and suggestions leading to improvement of the manuscript.

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
