# Peer review of "Retention time of lakes in the Larsemann Hills oasis, East Antarctica"

_The Cryosphere, 2020_

## Referee Comment (RC1) · Anonymous Referee #1 · 15 Oct 2020

L. 22: pro . . . ice sheets, shelf ices, glaciers. . . & . . . World. . . 27: . . . epishelf. . . 30: Leppäranta et al., 2020 is missing in References 34: . . . lasting for December–January (rather -February) 37: . . . are ice-free for a period of 3–4 months (rather 2-3 months) 70-71: Authors are citing that "85 % of precipitation falls as snow (World atlas, 1997)". 15 % of precipitation as rain is a strong overestimation not only for Larsemann Hills but for many if not all East Antarctic oases. Rain is extremely rare in East Antarctica.

80: Stüwe et al., 1989 is missing in References 80-81: However, Lake Reid was thermally and chemically stratidied also during open water period in Jan. 1994 (Kaup, E., Burgess, J.S. (2003). Natural and human impacted stratification in the lakes of the Larsemann Hills, Antarctica. In: Huiskes, A.H.L., Gieskes, W.W.C., Rozema, J., Schorno, R.M.L., van der Vries, S.M. & Wolff, W.J. (_EditorsAbbr). Antarctic Biology in

a Global Context (313−318).. Leiden, The Netherlands: Backhuys Publishers.

105: . . .14.5–15.0 m higher. . . - Sic!?

142: Using these SWE differences ignores ablation/sublimation of snow cover between the two surveys. Therefore the real water incomes are smaller and the LRTs respectively longer.

228: Still, data exist, f. ex. from the Schirmacher Oasis that ablation from lake's ice surface may reach 25 cm/year (Kaup, E., Haendel, D. (1995). Snow and ice cover of water bodies. In: P. Bormann & D. Fritsche (_Eds.). The Schirmacher Oasis, Queen Maud Land, East Antarctica (279−285). Gotha: Justus Perthes Verlag. (Petermanns Geographische Mitteilungen; 289

249: Interestingly, the level of Lake Nella remained stable f. ex. in summerss 1993/94 and 1997/98 until the first week of February after having reached the outflow threshold level at the end of December/beginning of January. There was no abrupt level drop these years. Obviously, the condition of abrupt level drop is the build-up of snow dam in previous winter.

272-73: Episodic hydrological observations on the lakes and streams located in the Schirmacher oasis (East Antarctica) date back to late 1980s. In fact, these 1983/84 runoff measurements for 6 lakes covered the full hydrological season from November until March.

277: Kaup 2002 and Loopmann & Klokov are missing in Reference List. In Kaup 2005 there are data that would result LRT in lakes Smirnova and Pomornik ca 1 and 2,4 years, respectively. In Lake Glubokoye (volume comparable to Lake Progress) the LRT would be 2.6 years.

371: . . .out in 1986–1987 to estimate. . . pro 1983–1984

372: Kaup, 2002 pro Kaup, 2005

---

## Author Comment (AC1) · 19 Oct 2020

The referee links comments to the text line, therefore we will follow them while answering.

L22: we agree, corrected.

L27: typo is corrected.

L30: the reference is added to the text.

L34: we agree, corrected.

L37: we agree, corrected.

[Figure]

L70-71: the text was corrected as follow: ... most of precipitation falls as snow (Word atlas, 1997). Rain is rarely observed over the continental coastal areas and ice-free oases."

L80: we corrected the list by putting the references in alphabetic order.

L80-81: to mention a phenomena of thermal stratification in Lake Reid we corrected the test as follow: "... Most of these lakes are well mixed during the summer seasons (Shevnina and Kourzeneva, 2017), and an exception is only Lake Reid, which has saline water. In this lake, the thermal stratification resistant to the katabatic winds of over 14 ms -1 is observed in January 1993 (Kaup and Burgess, 2003)."

L105: we would suppose such height range may occur due to various geodesic systems used to measure the elevation of lake tables/stages, however further discussion with Boronina et al., 2020 may help to understand the case.

L142: we agree, that the snow measurements do not give true values for the LRT because of various errors, and not accounting for the sublimation from the snow cover is among others.

L228: it is good to know that such data exist! We will address our next study to lakes located in the Schirmacher oasis.

L249: we agree, that the abrupt level drops in Lake Nella is happen due to melting of the snow dam formed in previous winter(s), and it is not necessary that such drops occur every year (Klokov, 1978).

L272-273 and L277: we corrected the text as follow: "... The hydrological observations on 6 lakes and streams located in the Shirmacher oasis (East Antarctica) date back to early 1980s. These observations cover whole hydrological season lasting from November, 1983 – March, 1984, and further they are used by Loopman and Klokov (1988) to estimate for... .... In these cases, the estimated LRT are 1 and 2.4 years for Lake Smirnova and Lake Pomornik correspondingly (Kaup, 2005). The volume

of these lakes is much less than the volume of Lake Nella/Scandrett/LH72 and Lake Progress/LH57. The LRT of Lake Glubokoe is estimated as 2.6 year, and it is almost three times less than for Lake Progress/LH57 which is comparable of volume..."

L371: corrected.

L372: corrected.

We thank our Anonymous Referee for the comments, suggestions, and a new (for us) knowledge on a phenomena of thermal stratification observed in the lakes located in the Larsemann Hills.

Elena Shevnina, from behalf of the authors

---

## Referee Comment (RC2) · Anonymous Referee #1 · 21 Oct 2020

L70-71: the text was corrected as follow: ... most of precipitation falls as snow (World atlas, 1997).

Isn't it still the WORLD Atlas?

L80-81: to mention a phenomena of thermal stratification in Lake Reid we corrected the test as follow: "... Most of these lakes are well mixed during the summer seasons (Shevnina and Kourzeneva, 2017), and an exception is only Lake Reid, which has saline water. In this lake, the thermal and salinity stratification resistant to the katabatic winds of over 14 ms -1 is observed in January 1993 (Kaup and Burgess, 2003)."

The water in Lake Reid can be considered brackish not really saline (Table 2 in Kaup and Burgess, 2003). Therefore instead of "... which has saline water." the reviwer

recommends "...which has BRACKISH water.

...observed in January 1993... It was in January 1994 (Kaup and Burgess, 2003).

The rest of answers are accepted by the reviewer.

---

## Author Comment (AC2) · 4 Nov 2020

We have implemented the following changes in the text of the manuscript: L70-71: we do not mention about a portion of snow fraction in the annual precipitation amount, therefore the reference to the Word Atlas is excluded from the text. L80-81: both corrections given by the reviewer are done.

---

## Referee Comment (RC3) · Anonymous Referee #1 · 5 Nov 2020

The Reviewer is satisfied with the Author's answers

---

## Referee Comment (RC4) · Martin Truffer (Referee) · 23 Nov 2020

[referee-annotated manuscript omitted]

---

## Author Comment (AC3) · 21 Dec 2020

Martin Truffer gives useful general comments in his review and provides detailed suggestions in order to improve the manuscript in the review supplement. Almost all the suggestions were implemented in the new version of the manuscript.

Further, we answer the general comments listed by Martin Truffer:

"1) The manuscript needs to be very carefully checked for language and grammar... " The text of the manuscript will be checked for typos, and in the final revised version will be prepared within next couple of weeks. We will English language will be also improved by a professional translator in order to fit the UK English standard. We have attached the re-considered text of the manuscript to the answer, and this version of the

manuscript is not yet checked.

"2) Results and Discussion should not be mixed into one section. . . " In the new version of the manuscript, Discussion section was separated from Results section. Discussion is now extended with the explanations: how our results fit to the past scientific records (lines 401-443); why the hydrological observations on the Antarctic lakes are important for a global scale prediction afterwards, specially in fast climate warming (lines 381-388 ); and what is the next step in the further study of water balance and thermal regime of the glacial lakes (lines 444-451).

"3) The Methods section would benefit from a table that shows which method was used for which lake. . . " We have included the new table (Table 2) in the section Methods in the new version of the manuscript. We also explain improved Figure 1 and Figure 2 (both given in the attachment) by given scale bar and legend.

"4) The numbers provided in the tables in 'Results' need to be provided with some amount of error estimates. . . " It is difficult to provide the numbers in the tables with the precise estimates of the errors inherent to them because this needs a separate study. The errors in the LRT include the uncertainties coming from measuring techniques and methods used to evaluate the terms of the water balance equation, as well as the surface area and volume of the lakes. However we include in the revised text of the manuscript the estimates of uncertainties for the water discharge measurements, area/ volume (lines 283-295) and evaporation (lines 328-333).

"5) The Conclusions should contain some sort of statement of what can be learned from these results in terms of how these lakes function. Do these retention times come as a surprise? Do they change the way we need to think about these lakes?

In the revised version of the manuscript, the section Conclusions is new, and it is includes answers to the questions: What is specific for water exchange for the epiglcial and land-locked lakes? (lines 463-469) What needs to be monitored on the lakes and streams? (lines 470-476)

We added the following references in the revised version of the manuscript:

Bell R., Banwell A., Trusel L., Kingslake J. Antarctic surface hydrology and impacts on the ice-sheet mass balance. Nature climate change, 2019, doi: 10.1038/s41558-018-0326-3.

Borgini F., Colacevich A., Loiselle S., Bargagi R., Short-term dynamics of physico-chemical and biological features in a shallow evaporative Antarctic lakes, Polar Biol, 36: 1147-1160, 2013, doi: 10.1007/s00300-013-1336-2

Kaup, E., and Burgess, J.S.: Natural and human impacted stratification in the lakes of the Larsemann Hills, Antarctica. In: Huiskes, A.H.L., Gieskes, W.W.C., Rozema, J.,Schorno, R.M.L., van der Vries, S.M. & Wolff, W.J.. Antarctic Biology in Global context, Leiden, The Netherlands: Backhuys Publishers, pp. 313-318, 2003.

Leppäranta, M., Luttinen, A., Arvola, L.: Physics and geochemistry of lakes in Vestfjella, Dronning Maud Land. Antarctic Science, 32(1), 29-42. doi:10.1017/S0954102019000555, 2020.

Popov, S.V., Sukhanova A.A., Polyakov, S.P.: Using georadar profiling techniques for the safety of transport operations of the Russian Antarctic Expedition // Meteorologiya I Gidrologiya, 2020, # 2, 126– 131 pp. [In Russian]

Shevnina E., Kourzeneva E., Potes M.: Evaporation over lakes of the Schirmacher oasis, East Antarctica. In book of abstracts "Complex investigation of the natural environment of the Arctic and Antarctica", St. Petersburg, Russia, 2-4 March, 2020, doi: 10.13140/RG.2.2.33613.38883

Zhelezhnjakov G. V., Danilevich B.B.: Accuracy of the hydrological measurements and estimations. Lenigrad, Gidrometeoizdat, 1966, 240 p. [In Russian]

We thank Martin Truffer for his corrections and suggestions, they helped to improve our manuscript a lot.

Elena Shevnina, on behalf of the authors

Please also note the supplement to this comment:
https://tc.copernicus.org/preprints/tc-2020-205/tc-2020-205-AC3-supplement.pdf

────────────────────────

[Figure]

[Figure]

**Fig. 1.**

[Figure]

**Fig. 2.**

**Supplement:**

[revised manuscript text omitted]
 takes from 2 to 15 years. The long water transport is common for the endoheic lakes since they loos water only by evaporation, which is expected be small in the cold polar environment. It should be noted that in this study we did not account for the sublimation from the lake surface during the ice covered period, although it could be a significant outflow term of the water balance for polar lakes (Huang et al., 2019; Faucher et al., 2019). This can lead to over-estimations of the LRT calculated from the outflow terms of the water balance equation both for the epiglacial and land-locked lakes.

By recent, it is little known about the water transport time scales of the glacial lakes located in the Antarctica, and it makes difficult to analyse our results comparing them with other studies. Foreman et al. (2004) apply the longest hydrological

observations to evaluate the hydraulic residence time of three large epiglacial lakes located in the Dry Valley, West Antarctica (Lake Bonney, Lake Hoare and Lake Fryxell). The estimated hydraulic residence time for Lake Bonney is 77 years, for Lake Hoare is 281 years and for Lake Fryxell is 107 years. The authors notice that the estimations of the residence time depend on the series length and period of the hydrological observations included into the analyses. In case of Lake Fryxell, for example, the hydraulic residence time varies from 107 to 9 years being calculated from data for years 1995–2001 and year 2001–2002, respectively. The volumes of these lakes are much bigger then of the lakes located in the Larsemann Hills oasis. All this makes the comparison between our estimations and ones given by Foreman et al. (2004) difficult.

The hydrological observations on six lakes and streams located in the Shirmacher oasis (East Antarctica) date back to early 1980s. These observations covered the whole hydrological season lasting from November, 1983 to March, 1984. Further they were used by Loopman and Klokov (1988) to estimate the water exchange coefficient (the inverse of the LRT) for three epiglacal lakes: Lake Smirnova with the volume of $613 \times 10^3$ m$^3$, Lake Pomornik with the volume of $236 \times 10^3$ m$^3$, and Lake Glubokoe with the volume of $1930 \times 10^3$ m$^3$. The estimated LRTs are 1 and 2.4 years for Lake Smirnova and Lake Pomornik correspondingly (Kaup, 2005). The volume of these lakes is much less than the volume of Lake Nella/Scandrett/LH72 and Lake Progress/LH57. The LRT of Lake Glubokoe, which is comparable in volume with Lake Progress/LH57, is estimated as 2.6 year, and it is almost three times less than for Lake Progress/LH57. It is interesting to note, that Lake Progress is connected with the epiglacial Lake Boulder, which got the outburst flood in January 2018 (Popov et al., 2020). It was the first observed outburst on this lake since start of instrumental observations in the Larseman Hills oasis in early 1990s.

The results show a big difference in the LRT estimated for the land-locked lakes Stepped/LH68, Sarah Tarn/LH71 and Reid/LH70. The LRT estimates depend much on the methods: for Lake Stepped/LH68, the LRT estimated from the outflow surface runoff is less than 2 years while from the inflow due to melting of seasonal snow cover is 13–30 years. For the endorheic Lake Sarah Tarn/LH71 and Lake Reid/LH70, the LRT estimated from the outflow term is also dependent on the method to calculate evaporation. If an empirical approach is used to calculate evaporation, the estimated LRT is smaller, it is 15 and 4 years for Lake Sarah Tarn/LH71 and Lake Reid/LH70, respectively. The LRT estimated from the inflow terms of the water balance is 34 years and 13 years for these lakes, respectively. We think that this difference is caused by the quality of the hydrological observations over snow. In particular, the length of the period and start/end dates depend on the schedule for seasonal logistic for each field campaign. We suggest to rely on the estimates from the outflow surface runoff since they are based on the hydrological measurements collected during longer periods (60–66 days). Data collected during the snow surveys in seasons 2011–2012 and 2016–2017 are not representative to estimate the inflow to the lakes due to melting of seasonal snow cover over their watersheds.

The LRT of three land-locked lakes vary from 2 to 15 years. The longest retention time is typical for the endoheic lakes loosing water with evaporation/sublimation. In this case, errors in the LRT estimates of depend on the methods used to

evaluate the evaporation. By recent, there are still only few studies addressing the evaporation over the Antarctic lakes. Most of them use the energy budget or semi-empirical equations to evaluate the evaporation from the meteorological observations (Dhote et al., 2020; Faucher et al., 2019). Borghini et al. (2013) evaluate the evaporation over three shallow land-locked lakes located the Victoria Land, West Antarctica. The surface area does not substantially decrease since late 1980s for these endorheic lakes. Authors suggest that water loss through surface evaporation is *ca.* 40–45 % of total change in the lake volume. In the Larsemann Hills, no significant changes in the surface area of the endorheic lakes Sarah Tarn and Reid are found since late 1980s (Shevnina and Kourzeneva, 2017), and the contribution of the evaporation may be significant  in the water balance of these lakes.

It should be noted that the direct measurements of the evaporation over the ice/snow and lakes are still rarely found for the remote polar regions, mostly due to specific measuring techniques. The traditional pan-evaporators are difficult to deploy and operate. The method of eddy covariance provides the best accurate estimates for the evaporation rate for the lakes. This method gives insight into the surface energy balance with the unique measurements, and allows improving the semi-empirical equations with best estimates of the regional coefficients (Tanny et al., 2011). We will address the future study  of evaporation over the epiglacial and land locked lakes with different methods: the eddy covariance, energy budget and semi-empirical. The method of eddy covariance will provide the reference while analysing the uncertainties in evaporation and the LRT estimates of the Antarctic lakes (Shevnina et al., 2020).

**5 Conclusion**

Water chemical composition and life presence in Antarctic lakes are strongly connected to the lake water balance and thermal regime. The modelling approach is among others approaches applied in understanding eutrophication, bio production and geochemical processes in lakes. Many geochemical models need characterization of water transport/exchange time scales. This study provides the first estimates for water transport time scale characteristic, namely the lake retention time  for five lakes located in the Larsemann Hills oasis, East Antarctica. The LRT was evaluated depending on a lake type: separately for the epiglacial and the land-locked lakes. The LRT was estimated with two methods based on the inflow and outflow terms of the water balance equation. The outflow terms are evaporation and surface outflow runoff; and the inflow terms are the water inflow due to melting of snow cover and precipitation over the lake surface. To calculate these components, we used the hydrological observations on the lakes and streams collected during four field campaigns in austral summers of 2011–2017.

Our study showed that the LRT of lakes in the Larsemann Hills oasis is very different, depending on the lake type. This is because the mechanisms of water exchange differ for the epiglacial and land-locked lakes. The epiglacial lakes loos water through the outflow surface runoff mostly. For these lakes, the surface runoff is much larger than the evaporation over the lake surface area. For the endorheic land-locked lakes, the evaporation over the surface area is the main loss term of the water balance equation. The obtained LRT indicates that water exchange in epiglacial lakes is faster than in land-locked endorheic lakes in the Larsemann Hills oasis, sometimes much faster. For example, the estimated LRT of large epiglacial

lake Nella/Scandrett/LH72 is 6–7 years, and of small land-locked lake Sarah Tarn/LH71 is 15-21 years. Therefore, different strategies should be applied to monitor the hydrological cycle of these lakes in general, and to obtain better estimates of the LRT in particular. For the epiglacial lakes, the accurate measurements on water discharges on inlet/outlet streams are needed, covering most of the warm season. If data are missing in the start or end of the warm period, over-estimation of the LRT may be expected. For the endorheic land-locked lakes, methods applied to measure/calculate evaporation are of great importance. All these methods need meteorological observations. To choose the best technique to calculate evaporation, measurement campaign with eddy covariance flux observations would be very helpful.  The role of sublimation over the ice-covered surface of the land-locked Antarctic lakes  should be also clarified. For these lakes, the LRT may be calculated also from the inflow term of the water balance equation, namely from snow melting. However snow surveys for that should start early enough, before the on-set of the melting season, and also end late enough. We recommend to carry out the first snow survey already in early December and the second one at the end of February. Otherwise, we may over-estimate the LRT a lot, as it happened in this study.

Hydrological observations on the epiglacial lakes also allow to accurately evaluate 
[revised manuscript text omitted]

---

## Author Response (AR1)

We thank Editor, Anonymous referee and Martin Truffer for the comments, suggestions, which were almost all implemented into the text of the revised manuscript. The expertise of both referees is helped to improve our manuscript a lot. Further, we listed the our answers to the comments and provided the information on the corrections done.

**Answers to the interactive comments given by the Anonymous referee**

The referee links the comments to the number of line (in the non-revised text), therefore we will follow them while answering.

Line 22: we agree, corrected.

Line 27: typo is corrected.

Line 30: the reference is added to the text.

Line 34: we agree, corrected.

Line 37: we agree, corrected.

Line 70-71: the text was corrected as follow: … most of precipitation falls as snow (Word atlas, 1997). Rain is rarely observed over the continental coastal areas and ice-free oases."

Line 80: we corrected the list by putting the references in alphabetic order.

Line 80-81: to mention a phenomena of thermal stratification in Lake Reid we corrected the test as follow: "… Most of these lakes are well mixed during the summer seasons (Shevnina and Kourzeneva, 2017), and an exception is only Lake Reid, which has saline water. In this lake, the thermal stratification resistant to the katabatic winds of over 14 ms$^{-1}$ is observed in January 1993 (Kaup and Burgess, 2003)."

Line 105: we would suppose such height range may occur due to various geodesic systems used to measure the elevation of lake tables/stages, however further discussion with Boronina et al., 2020 may help to understand the case.

Line 142: we agree, that the snow measurements do not give true values for the LRT because of various errors, and not accounting for the sublimation from the snow cover is among others.

Line 228: it is good to know that such data exist! We will address our next study to lakes located in the Schirmacher oasis.

Line 249: we agree, that the abrupt level drops in Lake Nella is happen due to melting of the snow dam formed in previous winter(s), and it is not necessary that such drops occur every year (Klokov, 1978).

L272-273 and L277: we corrected the text as follow: "… The hydrological observations on 6 lakes and streams located in the Shirmacher oasis (East Antarctica) date back to early 1980s. These observations cover whole hydrological season lasting from November, 1983 – March, 1984, and further they are used by Loopman and Klokov (1988) to estimate for…

…. In these cases, the estimated LRT are 1 and 2.4 years for Lake Smirnova and Lake Pomornik correspondingly (Kaup, 2005). The volume of these lakes is much less than the volume of Lake Nella/Scandrett/LH72 and Lake Progress/LH57. The LRT of Lake Glubokoe is estimated as 2.6 year, and it is almost three times less than for Lake Progress/LH57 which is comparable of volume..."

Line 371: corrected.

Line 372: corrected.

**Answers to interactive comments by Martin Truffer**

Martin Truffer gives useful general comments in his review and provides detailed suggestions in order to improve the manuscript in the review supplement. Almost all the suggestions were implemented in the revised version of the manuscript. Further, we answer the general comments listed by Martin Truffer:

"1) The manuscript needs to be very carefully checked for language and grammar… "

The text of the revised manuscript has been be checked for typos, and English language is improved by a professional translator in order to fit the UK English standard.

"2) Results and Discussion should not be mixed into one section… "

In the revised version of the manuscript, Discussion section is separated from Results section. Discussion is now extended with following explanations: how our results fit to the past scientific records (lines 401-443); why the hydrological observations on the Antarctic lakes are important for a global scale prediction afterwards, specially in fast climate warming (lines 381-388 ); and what is the next step in the further study of water balance and thermal regime of the glacial lakes (lines 444-451).

"3) The Methods section would benefit from a table that shows which method was used for which lake… "

We have included the new table (Table 2) in the section Methods in the new version of the manuscript. We also explain improved Figure 1 and Figure 2 (both given in the attachment) by given scale bar and legend.

"4) The numbers provided in the tables in 'Results' need to be provided with some amount of error estimates… "

It is difficult to provide the numbers in the tables with the precise estimates of the errors inherent to them because this needs a separate study. The errors in the LRT include the uncertainties coming from measuring techniques and methods used to evaluate the terms of the water balance equation, as well as the surface area and volume of the lakes. However we include in the revised text of the manuscript the estimates of uncertainties for the water discharge measurements, area/ volume (lines 283-295) and evaporation (lines 328-333).

"5) The Conclusions should contain some sort of statement of what can be learned from these results in terms of how these lakes function. Do these retention times come as a surprise? Do they change the way we need to think about these lakes?

In the revised version of the manuscript, the section Conclusions is new, and it is includes answers to the questions: What is specific for water exchange for the epiglacial and land-locked lakes? (lines 463-469) What needs to be monitored on the lakes and streams? (lines 470-476)

We also corrected the figures 1 and 2 by given the scale bar instead of the scaling factor, and defined abbreviations of the lake designations in the text after these two figures.

The following references were included to the revised version of the manuscript:

1. Bell R., Banwell A., Trusel L., Kingslake J. Antarctic surface hydrology and impacts on the ice-sheet mass balance. Nature climate change, 2019, doi: 10.1038/s41558-018-0326-3.

2. Borgini F., Colacevich A., Loiselle S., Bargagi R., Short-term dynamics of physico-chemical and biological features in a shallow evaporative Antarctic lakes, Polar Biol, 36: 1147-1160, 2013, doi: 10.1007/s00300-013-1336-2

3. Kaup, E., and Burgess, J.S.: Natural and human impacted stratification in the lakes ofthe Larsemann Hills, Antarctica. In: Huiskes, A.H.L., Gieskes, W.W.C., Rozema, J.,Schorno, R.M.L., van der Vries, S.M. & Wolff, W.J.. Antarctic Biology in Global context, Leiden, The Netherlands: Backhuys Publishers, pp. 313-318, 2003.

4.  Leppäranta, M., Luttinen, A., Arvola, L.: Physics and geochemistry of lakes in Vestfjella, Dronning Maud Land. Antarctic Science, 32(1), 29-42. doi:10.1017/S0954102019000555, 2020.

5.  Popov, S.V., Sukhanova A.A., Polyakov, S.P.: Using georadar profiling techniques for the safety of transport operations of the Russian Antarctic Expedition // Meteorologiya I Gidrologiya, 2020, # 2, 126– 131 pp. [In Russian]

6.  Shevnina E., Kourzeneva E., Potes M.: Evaporation over lakes of the Schirmacher oasis, East Antarctica. In book of abstracts "Complex investigation of the natural environment of the Arctic and Antarctica", St. Petersburg, Russia, 2-4 March, 2020, doi: 10.13140/RG.2.2.33613.38883

7.  Zhelezhnjakov G. V., Danilevich B.B.: Accuracy of the hydrological measurements and estimations. Lenigrad, Gidrometeoizdat, 1966, 240 p. [In Russian]

Elena Shevnina,

on behalf of the authors

---

## Referee Report (RR1)

[referee-annotated manuscript omitted]

---

## Author Response (AR2)

We thank Enn Kaup and Martin Truffer for their comments. They were almost all implemented into the text of the revised manuscript. Further, we listed the our answers to the comments and provided the information on the corrections done.

**Answers to the specific comments given by Enn Kaup:**

_1. In the Introduction the purpose of the study should be indicated more clearly (f. ex. also to point out that it's the first such one in the Larsemann Hills)._

"This study aims to evaluate the lake retention time of the lakes located in the Larsemann Hills oasis. We suggested to estimate the LRT from the outflow and inflow terms of the water balance equation depending on a type of lake (epiglacial and land-locked). ... This study gives the first estimations of the LRT for the lakes located in the Larsemann Hills." (ll. 56-57).

_2. In section 2.1 elaborate a bit more on sublimation/ablation of snowpack and on it's influence on LRT._

We extended speculation on the errors connected to sublimation as following: "In this study, we neglected to the water sublimation over the ice covered parts of the epiglacial lakes and snow packs. It is assumed that, in summer it is small compare to others components of the lake water balance equation. However, to proof the assumption would need to a separate study. Not accounting of the water sublimation may lead to slight over- estimation of the LRT, especially for the land-locked lakes. (ll. 158-167).

_3. In section 2.2. it needs to be recognized the substantial sublimation of lake ice cover during all the year and mention it in the Conclusions._

To stress the substantial role of sublimation, we added some speculation in the section 2.2, as well as extend discussion section:

L. 200: "... water sublimation over ice covered surface of a lake ... "

Ll. 268-269: "Even it may be a substantial (Kaup and Haendel, 1995), the evaporation during the frozen period (water sublimation) was also not accounted because no observations were available for its evaluations."

Ll. 463-465: "Since the water sublimation over ice covered lakes depends on meteorological conditions (air temperature and humidity, wind speed) which are largely similar in the East Antarctic oases, our next study address to evaluation of water loss for ice free and ice covered lakes. We would expect that the over- estimation of the LRT due to neglected sublimation would not exceed 10 % (Shevnina et al., 2020)."

Ll. 490-493: "The role of the water sublimation over the ice-covered surface of the lakes should also be studied in details. It may be substantial component of a lake water balance equation integrated over the period of year. "

_4. Fig. 1 needs to be complemented (shown in text)._

We added the missing parts of the catchment area for two epiglacial lakes (Progress and Nella/Scandrett).

_5. The references must be put correct (missing in the Reference list and vice versa)_

We put the references in alphabetic order, and added new references.

Loopman A., Klokov V.: The formation of water runoff from the lake catchments of the Schirmacher oasis in East Antarctica during the summer season 1983 – 1984, in Limnological studies in Quin Maud Land (East Antarctic), Ed. by J. Martin (Valgus, Tallinn), 57–65, 1988.

Simonov, I.M., and Fedotov, V.I.: Ozera oasisa Schimachera. [Lakes of the Schirmacher oasis]. Informazionny bulleten Sovetskoy Antarctichesko Expedicii, 47, 19–23, 1964. [In Rissian].

Vershinin K., Shevnina E.: Technical report of the hydrological studies in the Larsemann Hills in the season 58 RAE, in: Series of reports of the Russian Antarctic Research Expedition (RAE), 45 p, 2013. [in Russian]

Kaup, E., Haendel, D.: Snow and ice cover of water bodies. In: P. Bormann and D. Fritsche (Eds). The Schirmacher Oasis, Queen Maud Land, East Antarctica, Gotha: Justus Perthes Verlag, 279−285, 1995.

Hermichen, W. D., P. Kowski, and U. Wand: LakeUntersee, a first isotope study of the largest freshwater lake in the interior of East Antarctica. Nature,315,131–133, doi: 10.1038/315131a0, 1985.

Tanny, J., Cohen, S. , Assouline S., Lange F., Grava A. , Berger D. , Teltch B. , ParlangeM.B. 2008: Evaporation from a small water reservoir: Direct measurements and estimates. Journal of Hydrology, 351, 218– 229, doi: https://doi.org/10.1016/j.jhydrol.2007.12.012

Shevnina, E., Kourzeneva, E., and Potes, M.: Evaporation over lakes of the Schirmacher oasis, East Antarctica, In proceedings of International Conference "Complex study of the Arctic and Antarctica", March 2020, St. Petersburg, Russia, doi: 10.13140/RG.2.2.33613.38883, 2020.

We also added other changes in the text according the comments given in the text:

Ll. 37-40: "Small land-locked lakes are fully ice-free for a period of 2–3 months in summer (Lakes Sarah Tarn and Lake Reid in Larsemann Hills, Lake Verhnee in Shrimacher oasis). Big land-locked lakes can stay partially ice covered in summer, and a number of such lakes are found in the Schimacher oasis. Thalla Hills and Bunger Hills (Gibson et al., 2002; Loopman et al., 1988; Simonov and Fedotov, 1964).The land-locked lakes lose water mainly through the surface runoff in the outlet streams, and/or through evaporation over their surface."

L53: corrected as following: "...  lakes located in the Antarctic Dry Valleys, and ..."

L53: Loopman and Klokov, 1988 was added to the list of references.

L.125: Vershinin and Shevnina, 2013 is now added to the list of references.

Ll.152-153: corrected as following: … "Using these SWE differences may ignores water sublimation of snow cover between the two surveys, it will lead to over- estimation of the LRT for the lakes. "

L. 233: corrected as "… Bunger ..."

L. 272: added the reference to Kaup and Haendel (1995) into the text, and we also included information on the measured water sublimation in the section of discussion.

L. 414: replaced to "... East … "

L. 421: corrected to "…  Schirmacher … "

L. 444-445: corrected as "... Also, water loss through sublimation over the ice/snow cover in the catchment area can be substantial (Hermichen et al., 1985)… "

L. 451: corrected to "… East ..."

L. 460 we added Tanny et al., 2008 in the list of references.

**Answers to the specific comments given by Martin Truffer:**

*1. l.23: This statement could be a bit controversial: Just because all the water is frozen, that doesn't make the continent more sensitive to warming (compared to Greenland for example)*

We have replaced the statement with "Climate warming enhances melting of the ice sheets and glaciers, and melted water accumulates in lakes and streams." (ll. 22-25).

*2. Table 2: state the units (years) for LRT*

The unit for the LRT was added to the table2. (l.267)

*3. l.371: leaded -> led*

We corrected the text. (l. 382)

*4. l.484: it strikes me that the question 'how rapidly is water renewing in Antarctic lakes" is a somewhat odd question in the sense that the answer will surely be different, depending on size of the drainage basin and topography of the lake. For example, there is no good a-priori reason why two neighboring lakes should have similar LRTs*

We have replaced the text with the statement: "We would consider our LRT estimates as a preliminary attempt suggest two versions of the water balance equation depending on a lake types namely for the epiglacial and land-locked lakes." (ll. 501-502).

Elena Shevnina,
on behalf of the authors

---

## Author Response (AR3)

Dear Tom Shatwell,

Thank for all correction and suggestions, we have implemented all of them in the corrected text of the manuscript. We appreciate for all comments and questions made by the editor and two reviewers, they allow to substantial improvement our manuscript.

with the best regards,
Elena Shevnina

on behalf of authors